# Use of Glycoproteins—Prostate-Specific Membrane Antigen and Galectin-3 as Primary Tumor Markers and Therapeutic Targets in the Management of Metastatic Prostate Cancer

**DOI:** 10.3390/cancers14112704

**Published:** 2022-05-30

**Authors:** Satish Sharma, Katherine Cwiklinski, Donald E. Sykes, Supriya D. Mahajan, Kent Chevli, Stanley A. Schwartz, Ravikumar Aalinkeel

**Affiliations:** 1Division of Allergy, Immunology and Rheumatology, Department of Medicine, Jacobs School of Medicine and Biomedical Sciences, University at Buffalo, Clinical and Translational Research Center, 875 Ellicott St., Buffalo, NY 14203, USA; ss466@buffalo.edu (S.S.); kcwiklin@buffalo.edu (K.C.); desykes@buffalo.edu (D.E.S.); smahajan@buffalo.edu (S.D.M.); sasimmun@buffalo.edu (S.A.S.); 2Department of Urology, Jacobs School of Medicine and Biomedical Sciences, University at Buffalo, Buffalo, NY 14203, USA; chevli@buffalo.edu

**Keywords:** galectins, glycoproteins, prostate specific membrane antigen, metastasis, prostate cancer, tumor microenvironment

## Abstract

**Simple Summary:**

Prostate specific membrane antigen and galectins are proteins expressed on cell surface and their expression is associated with cancer growth and spread. The goal of this research was to look at the pattern of these two glycoproteins in the human prostate cancer microenvironment. Prostate specific membrane antigen and galectins-1,3 and 8 were the most frequently detected glycoproteins in various phases of this disease. Furthermore, prostate specific membrane antigen and galectin-3 expression are good indicators of tumor aggressiveness, and their combined expression can be valuable tool for prostate cancer diagnosis and treatment in future. Together, our findings reveal a tightly regulated “Prostate specific membrane antigen-galectin-pattern” that accompanies disease in prostate cancer and point to a key role for combined prostate specific membrane antigen and galectin-3 inhibitors in prostate cancer treatment along with standard chemotherapy.

**Abstract:**

Galectins and prostate specific membrane antigen (PSMA) are glycoproteins that are functionally implicated in prostate cancer (CaP). We undertook this study to analyze the “PSMA-galectin pattern” of the human CaP microenvironment with the overarching goal of selecting novel-molecular targets for prognostic and therapeutic purposes. We examined CaP cells and biopsy samples representing different stages of the disease and found that PSMA, Gal-1, Gal-3, and Gal-8 are the most abundantly expressed glycoproteins. In contrast, other galectins such as Gal-2, 4–7, 9–13, were uniformly expressed at lower levels across all cell lines. However, biopsy samples showed markedly higher expression of PSMA, Gal-1 and Gal-3. Independently PSA and Gleason score at diagnosis correlated with the expression of PSMA, Gal-3. Additionally, the combined index of PSMA and Gal-3 expression positively correlated with Gleason score and was a better predictor of tumor aggressiveness. Together, our results recognize a tightly regulated “PSMA-galectin- pattern” that accompanies disease in CaP and highlight a major role for the combined PSMA and Gal-3 inhibitors along with standard chemotherapy for prostate cancer treatment. Inhibitor combination studies show enzalutamide (ENZ), 2-phosphonomethyl pentanedioic acid (2-PMPA), and GB1107 as highly cytotoxic for LNCaP and LNCaP-KD cells, while Docetaxel (DOC) + GB1107 show greater efficacy in PC-3 cells. Overall, 2-PMPA and GB1107 demonstrate synergistic cytotoxic effects with ENZ and DOC in various CaP cell lines.

## 1. Introduction

Prostate cancer (CaP) is the third most commonly diagnosed cancer in men and remains one of the most frequent causes of cancer-related deaths among males [1]. Management of the metastatic CaP remains challenging. Early diagnosis and treatment are paramount to improving the treatment outcome. The prostate-specific antigen (PSA), a glycoprotein enzyme abundantly expressed in prostate tissue continues to be the most widely used marker for diagnosis and prognostic assessment in prostate cancer patients’ post-treatment. Many recent studies have demonstrated that PSA levels in patients with metastatic CaP are not linearly related to prognosis [2]. Other options for diagnosis such as magnetic resonance imaging (MRI), positron emission tomography (PET), computerized tomography (CT), or bone scans are expensive and limited [3].

The optimal diagnosis and management of CaP has therefore shifted to certain accurate biomarkers to stratify patients with increased possibility of harboring aggressive tumors. One such biomarker is prostate-specific membrane antigen (PSMA), also known as folate hydrolase-1. PSMA is a transmembrane glycoprotein initially identified in CaP cell-line LNCaP [4,5]. In normal prostate cells, PSMA resides in the cytoplasm. However, through tissue breaches following malignant transformation, it relocates to have substantial extracellular presence. Relocated PSMA can be targeted for both diagnostic and therapeutic purposes [6]. It has been reported that PSMA is present at a low level even in normal prostate tissue, but the amount is markedly higher in CaP cells. PSMA expression has also been reliably demonstrated in normal and hyperplastic prostate tissue, prostatic intraepithelial neoplasia, and invasive carcinomas using immunohistochemistry (IHC) and other techniques [7,8,9]. In CaP specimens, PSMA expression and enzymatic activity are higher than in benign prostate tissue and are linked to high tumor grade and the development of metastases [9,10], implying that PSMA expression is a bad prognostic indicator of the illness.

Additionally, galectins, a family of 14 glycan-binding proteins, another group of glycoproteins affect every stage of tumor progression. In recent studies, galectins have been identified as vital elements in the CaP microenvironment [11,12]. A member of the galectin family, galectin-1 (Gal-1), regulates CaP cell differentiation, survival [13], and inhibition of T-cell transmigration [14]. Another member of this family, galectin-3 (Gal-3), is a tumor-associated protein present in the seminal fluid and is a substrate for the PSA enzyme [15]. Experimental and clinical data from different studies show that Gal-3 levels in the serum of patients with metastatic CaP are persistently elevated compared to the controls without CaP [16]. Alternatively, Gal-3 is believed to regulate the aggregation of CaP cells in vitro [17,18]. Moreover, Gal-3 expression in CaP cells has been proposed as an indicator of the shift from benign to castration-resistant disease [19], and its controlled expression has been linked to promoter methylation [20].

However, despite significant advancement in understanding the role of individual glycoproteins and PSMA, there are no data available on the role played by the co-existence of altered PSMA and/or galectin expression in different stages of progression and metastasis of CaP. Therefore, we conducted this study to see if there is a link between the expression of different galectins and PSMA, in CaP cell lines and in prostate biopsy and to identify if there is a unique galectin/PSMA signature that could be targeted for combination therapy with the overarching goal of selecting novel-molecular targets for prognostic and therapeutic purposes.

## 2. Material and Methods

### 2.1. Normal Prostate and CaP Cell Lines

Normal prostate epithelial cells PWR-1E and the remaining CaP cells- PC-3, DU-145, and LNCaP cells were all obtained from the American Type Culture Collection (Manassas, VA, USA). PWR-1E cells are primary normal epithelial cells obtained from a non-neoplastic adult human prostate and infected with the Ad12-SV40 virus and express markers of normal prostatic epithelial cells and mimic normal growth and differentiation responses to androgens. On the other hand, the different CaP cells used here differ significantly in their aggressive phenotypes, as demonstrated by us previously, with PC-3 > DU-145 > LNCaP [21]. PC-3 is a cell line which was established from metastases in lumbar vertebrae of a prostate cancer patient and is androgen independent. DU-145 was isolated from metastases in the brain and is androgen receptor (AR) positive. LNCaP is AR positive and androgen responsive but growth does not depend upon androgen. LNCaP-KD is PSMA knock down cell line established in our laboratory using the CRISPR/cas9 system to disrupt the PSMA gene in LNCaP cells. All cell lines were cultured in RPMI 1640 medium supplemented with 10% heat-inactivated fetal bovine serum 2 mmol/L l-glutamine, 100 µg/mL streptomycin, and 100 U/mL penicillin, a two-fold vitamin solution (Life Technologies, Grand Island, NY, USA) and grown at 37° in a humidified atmosphere of 95% air and 5% CO_2_. Cells were harvested by trypsinization for use in routine experiments. 

### 2.2. Gene Expression for Galectins and PSMA in CaP Cells

For gene expression analysis, cytoplasmic RNA was extracted from different cells by an acid guanidinium-thiocyanate-phenol-chloroform method, using the TRIzol^®^ reagent (Invitrogen, Grand Island, NY, USA). The final RNA pellet was dried and resuspended in diethyl pyrocarbonate-treated water and the concentration of RNA was determined by spectrophotometry at 260 nm. Relative expression of mRNAs of interest was assessed using the SYBR green master mix from Stratagene (La Jolla, CA, USA) to perform quantitative PCR (qPCR) using the Stratagene MX3005B (La Jolla, CA, USA). Differences in the threshold cycle number are used to quantify the relative amount of PCR target contained within each tube. The relative mRNA expression is quantified as transcript accumulation index, calculated using the comparative *C*_T_ method. All data were controlled for the quantity of RNA input by performing measurements on a reference gene, β-actin and expressed as fold increase compared to the expression of the keeping gene.

### 2.3. Biopsy Samples

Prostate cancer biopsies were obtained from the archived tissue bank of the Department of Urology at the Great Lakes Cancer Center Cheektowaga, NY. Gleason score and Tumor grading was performed without knowledge of the PSMA and galectin results. Immunoreactivity was scored for fluorescence intensity using ImageJ analysis and results were correlated with pre-treatment PSA level and Gleason score. Specimens (*n* = 115) covered different Gleason grade and controls (Table 1). None of the patients received any preoperative therapy. Protocols were approved by the Ethics Committee of the Jacobs School of Medicine and Biomedical sciences, Buffalo, NY, USA.

### 2.4. Immunofluorescence (IFC)

IFC was performed on paraffin-embedded biopsy samples as described below. Initially the samples were deparaffinized by 5-min incubation sequentially in xylene, 100%, 95%, and 75% ethanol in that order. Nonspecific binding was blocked using normal horse serum in 0.05% saponin. This was followed by incubation with the corresponding antibodies primary antibodies, PSMA (monoclonal anti-PSMA antibody, clone 3E6, Dako) Gal-8; (Recombinant Anti-galectin 8/antibody, Cat # ab109519 AbCam), Gal-3; (Anti-galectin-3 Antibody, Cat # sc-53127, Santa Cruz Biotechnology), Gal-1 (Anti-galectin-1 Antibody, Cat # sc-166618, Santa Cruz Biotechnology). The secondary antibodies used include Alexa Fluor^®^ 488 Anti-Rabbit Secondary Antibodies, and Alexa Fluor^®^ 594 rabbit anti-mouse IgG, all obtained from Thermo Fisher Scientific, Grand Island, NY, USA. DAPI (Thermo Fisher Scientific) was used to stain nuclei. For IFC in CaP cells, cells were grown to 70% confluence in glass petri dishes and fixed for 10 min at 37° in 4% formaldehyde, followed by permeabilization with ice-cold 90% methanol. Cells were washed in 1× PBS and treated with primary antibodies against PSMA, Gal-1, Gal-3, and Gal-8 followed by staining with cy5-labelled goat anti-rabbit antibodies (Molecular Probes, Grand Island, NY, USA) to determine the expression levels of these proteins. The expression level of galectin-1, galectin-3, gaectinl-8 and PSMA were quantitated using fluorescence imaging done using the EVOS^®^ FL Cell Imaging System (Life Technologies, Grand Island, NY, USA). The intensity of the fluorescent signal was quantitated using the ImageJ software (National Institutes of Health, Bethesda, MA, USA).

### 2.5. Western Blots for Different Galectins and PSMA

Cell lysates were prepared by re-suspending cells in lysis buffer [65 mmol/lTris–HCl (pH 7.4), 150 mmol/L NaCl, 1 mmol/L EDTA, 1% nonidet-P40, 1% sodium deoxycholate, 1 μg/mL aprotinin, 100 μg/mL PMSF, for 30 min at 4° and cleared by centrifugation for 30 min at 13,000× *g*. Supernates were collected and total protein concentration was determined using Coomassie Protein Reagent (Bio-Rad, Hercules, CA, USA), fractionated by 10% SDS–PAGE, transferred to nitrocellulose membranes, incubated with primary antibody Gal-8 (Anti-galectin-8 Antibody: JB85-35 from Novas Biologicals used at 1:100), Gal-3 (Human Galectin-3 Antibody (Cat # A3A12, abcam, used at 1:100) Gal-1 (Anti-galectin-1 Antibody (C-8): sc-166618 from Santacruz Biotech used at 1:100) and PSMA (Anti-PSMA antibody (aa161-190) (Cat # LS-C166202-400) from LSBio, Seattle, WA, used at 1:100) and horseradish peroxidase-conjugated secondary antibodies, and revealed with Super Signal Pico West (Pierce, Rockford, IL, USA). β-Actin expression was used as an internal control.

### 2.6. Flow Cytometry

Flow cytometry was performed for different galectins in LNCaP, DU-145, and PC-3 by using ~1 × 10^6^ cells from each cell lines. The cells were initially suspended in FACS buffer (1× PBS, 2% calf serum, 5 mM EDTA, 0.1% sodium azide) followed by staining of cells with antibodies against Gal-8 (Anti-galectin-8 Antibody (C-8): sc-377133 from Santacruz Biotech used at 1:100), Gal-3 (Human Galectin-3 Antibody Cat # MAB11542, Bio-Techne Corporation used at 1:100), Gal-1 (Anti-galectin-1 Antibody (C-8): sc-166618 from Santacruz Biotech used at 1:100), and PSMA (Anti-PSMA antibody (GCP-05) (Cat # ab66912) from abcam used at 1:100) followed by labelling with Alexa Fluor 488 labeled secondary mouse antibody (Jackson Immuno Research (Cat # 209-545-082 at 1:1000). Flow cytometry was performed with a BD Biosciences LSR II where 10,000 cells were gated and data processing was performed with FlowJo version 7.6 software (Tree Star, Ashland, OR, USA).

### 2.7. Gal-3 and PSMA Inhibitor Studies

Using three different CaP cell lines, PWR-1E, LNCaP, LNCaP KD (PSMA knock down), and PC-3, the effects of PSMA-inhibitor (2-PMPA, Sigma-Aldrich, St. Louis, MO, USA), Gal-3 inhibitor (GB1107, MedChemExpress, Monmouth Junction, NJ, USA), Androgen receptor (AR) blocker (Enzalutamide, Selleck Chemicals, Houston, TX, USA), and microtubule destabilizer (Docetaxel (Doc), LC Laboratories, Woburn, MA, USA) either alone or in combination were analyzed. About ~2.5 × 10^3^ cells from each cell line were grown overnight in a 96-well plate containing complete growth media (RPMI 1640), following which the cells were treated with required concentrations of the drug(s) either independently or in combination. After 72 h of exposure to the drugs either alone or in combination, the viability of surviving cells was determined by the 3 (4,5 dimethylthiazol 2 Yl) 2,5 diphenyltetrazolium bromide (MTT) assay using a microtiter plate reader (Bio Tek Instruments, Inc., Winooski, VT, USA) at 570 nm. Cell proliferation/inhibition was calculated as the percentage of proliferation/inhibition: [1 − (A/B)] × 100, where A is the absorbance of treated cells, and B is the absorbance of untreated control cells expressed as % of control cells. Finally, a comparative analysis was performed to identify the effectiveness of each treatment alone and together on CaP cell viability to determine therapeutic application.

### 2.8. Statistical Analysis

All experiments were repeated at least four times in triplicate. Values are expressed as the mean ± SD. The significance of the difference between the control and each experimental test is analyzed by unpaired *t*-test (GraphPad, Prism 9.0) and a value of *p* < 0.05 was considered statistically significant. For correlational analysis χ^2^ test was used to determine the significance of the associations between Gleason score and expression of variables.

## 3. Results

### 3.1. PSMA and Galectin Expression Pattern

To define the galectin and PSMA expression patterns associated with CaP growth, we first looked at their transcriptional expression in different human CaP cell lines representing distinct stages of the disease. We have earlier reported that these CaP cell lines have various aggressive/metastatic activities that are linked to pro-angiogenic factor gene expression and protein synthesis [21]. Our report was the first concrete proof of a relationship between CaP cells’ metastatic potential and their expression of pro-angiogenic cytokines. In this study, we used the same CaP cell lines, the hormone-responsive LNCaP, AR-positive (DU-145), and AR-negative (PC-3) castration-resistant cell lines and quantified the gene and protein expression of different galectins and PSMA. Total RNA was collected and examined using quantitative RT-PCR in the log phase of growth (Figure 1A). The gene expression of PSMA and galectins showed different patterns between low aggressive LNCaP cells, the castration-resistant moderately aggressive DU-145 cells, and the highly aggressive PC-3 cells. Results of our study have shown that the transcripts of PSMA were expressed ~47 fold higher in the least aggressive LNCaP cells, whereas the moderately aggressive DU-145 and the highly aggressive PC-3 cells did not show detectable transcript accumulation of PSMA. When the transcript level of different galectins was assessed, we found the accumulation of Gal-1 in all three cells, with LNCaP showing a roughly ~29-fold increase and the aggressive DU-145 and PC-3 cells showing a remarkable ~400-fold increase. However, Gal-3 transcript accumulation was observed only in DU-145 and PC-3 cells ~110-fold and ~200-fold respectively. Results of our study also show transcript accumulation for Gal-8, which is a prostate cancer marker [22,23], expressed across all CaP cell lines studied, with LNCaP cells showing the highest expression (~128-fold). All other galectin family members (Gal-2,4,5,6,7,9,10,12,13, and 14) had low transcript levels. Further, we assessed the expression of galectin family members at the protein level to better define the “Prostate cancer signature based on galectins” (focusing only on galectins with higher transcript abundance). Our protein analysis shows a similar pattern as that observed in the gene transcript-level studies with Gal-8 being copiously expressed galectin in the least aggressive LNCaP cell lines (Figure 1B) and Gal-1 and Gal-3 protein expression detected only in DU-145 and PC-3 cell line. Other galectins had negligible to minor expression in all of the cell lines studied (data not shown). These data suggest that galectin expression is tightly regulated in CaP cell lines with Gal-8 being highly expressed in the least aggressive LNCaP cell line and Gal-1 and Gal-3 highly expressed in aggressive cell lines DU-145 and PC-3. 

We extended our analysis using Western blot and flow cytometry to corroborate the results obtained by Q-PCR for different galectins and PSMA to confirm if observed gene expression resulted in increased translation of the gene product (Figure 2). Initially, we performed Western blotting followed by flow cytometry analyses. For both analysis we selected only those galectins and PSMA, which had shown a maximal and differential expression in our qPCR analysis. The results depicted in Figure 2 show that the translated protein products showing similar patterns at the protein level as observed in our qPCR analysis. The products of the β-actin, Gal-8, Gal-3, Gal-1, and PSAM genes were detected as predicted 42, 48, 72, 14, and 72 -Kd fragments, respectively, by the specific antibodies used (Figure 2A). All bands were analyzed by densitometry and the values were normalized to the values of the constitutively synthesized, β-actin (Figure 2B). As observed in gene expression studies we found LNCaP cells to express the highest level of Gal-8. Whereas Gal-3 and Gal-1 proteins were several folds higher in DU-145 and PC-3 compared to low levels in LNCaP cells. As predicted, PC-3 and DU-145 cells did not express any detectable levels of PSMA, whereas LNCaP cells showed ~10-fold increase in the translational level of this protein. 

Further validation of our immunofluorescence data on the expression of selected galectins was performed by flow cytometry analysis. Flow cytometry analysis shows the specific binding of each antibody with each cancer cell, type LNCaP, DU145, and PC-3 respectively (Figure 2C). Results of the flow cytometry analysis show a similar pattern of expression of Gal-8 with LNCaP cells expressing 96% followed by DU-145 at 34.4% and PC-3 at 26.8% (Figure 2C top panel). Gal-1 expression by flow cytometry shows this galectin to be least expressed in LNCaP cells (24.1%) and higher expression in both DU-145 (83.9%) and PC-3 cells (83.8% Figure 2C 2nd panel). Baseline expression of Gal-3 by flow analysis shows minimal expression in the least aggressive LNCaP cells (38.9%) and higher expression in DU-145 and PC-3, at 94.7 and 89.7% respectively. The results in Figure 2C also show that PSMA expression was the highest in LNCaP (79.9%) and negligible expression in DU-145 and PC-3 cells (2.37% and PC-2.31% respectively). The expression levels were represented with respect to the unstained cells as control. These data, along with our Western blot data in Figure 2A, corroborate our observation from immunofluorescence staining and further validate the conclusions based on our immunofluorescence study.

### 3.2. Gal-1, Gal-3, and PSMA Protein Expression in Prostate Biopsies

Armed with information about the expression pattern of galectins and PSMA in different CaP cell lines, we then investigated if this observation is translatable in vivo. For this purpose, we quantified the expression of PSMA, Gal-1&3 in a set of 115 biopsy specimens (105 CaP^+^ and 10 CaP^−^ by IFC (Figure 3). The biopsy samples differed in a wide range in their Gleason score. Our results show that CaP^+^ biopsies expressed a ~31-fold (range 7.6–87-fold) increase in PSMA and a ~17-fold (2.3–35-fold) increase in Gal-3 compared to control biopsies. However, Gal-1 only showed ~1.5-fold increase in CaP^+^ biopsies compared to CaP^−^ samples. Other galectins were not quantified in the biopsy samples. In univariate analysis, there was a significant and strong correlation between PSMA pixel unit vs. Gal-3-pixel unit, (r^2^ = 0.753, *p* < 0.001, Figure 3B(I). However, there was no such correlation with the levels of galectin-1 pixel unit. On further analysis, higher expression of PSMA correlated with high Gleason score grade (r^2^ = 0.597, *p <* 0.001, Figure 3B(II)) and a week correlation between Gal-3 pixel units and Gleason score (Figure 3B, r^2^ = 0.434, *p* < 0.001, Figure 3B(III)). However, when the levels of PSMA and Gal-3 are added, the combination intensity in pixel units showed a strong correlation with Gleason score (r^2^ = 0.713, *p* < 0.001, Figure 3B(IV). The mean serum PSA level of 183.85 ng/mL (Table 1) at the time of diagnosis for the PSMA-overexpressing tumors was significantly greater than the mean serum PSA level of 3.25 ng/mL for the PSMA-non expressing group (*p* < 0.001). The mean Gleason score of tumors within CaP^−^ samples with minimum background PSMA expression was 5, and the mean Gleason score of the tumors with PSMA overexpression was 8.0.

### 3.3. Combined Targeting of PSMA and Gal-3 for a Therapeutic Advantage in Management of Prostate Cancer

With increased understanding of CaP biology, combination treatments targeting the androgen receptor (AR) and phosphoinositide 3-kinase (PI3K) pathways have been proposed for treating advanced CaP. However, with the knowledge gained through this study showing increased expression of Gal-3 and PSMA in aggressive CaP and with the prevailing knowledge on the functional role of these two glycoproteins in proliferation and aggressiveness, we hypothesize that targeting these two glycoproteins with corresponding inhibitors will be an effective strategy in the management of aggressive CaP. Accordingly, we selected inhibitors of PSMA (2-PMPA) and Gal-3 (GB1107) that have shown effectiveness in inhibiting their respective targets for their application either independently or conjointly in the treatment of prostate cancer [24,25]. Our results demonstrate that enzalutamide (ENZ) alone showed comparable inhibition in both LNCaP and LNCaP-KD, with ~80% cell death at the highest dose (*p* < 0.001) but was ineffective in producing any detectable cell death in PC-3 cells (Figure 4A). Additionally, the exposure of LNCaP cells to 2-PMPA showed a dose-dependent effect with ~52% cell death at 200 nM (*p* < 0.001) and no quantifiable cell death in LNCaP-KD and PC-3 cells (Figure 4B). However, GB1107 by itself showed dose-dependent cytotoxic effect on all cell lines with ~60% cell death at 10 μM, the highest dose (Figure 4C). We also independently verified the effects of each drug on PWR-1E normal prostate epithelia cells. The results of this study (Figure 4D) show that of the drugs, only ENZ showed a dose-dependent effect on PWR-1E cells. We observed around 87% cell death at the highest dose of the drug tested. The other two drugs 2-PMPA and GB-1107, did not show any detectable cytotoxic effect on PWR-1E cells.

Combination treatment of LNCaP and LNCaP-KD with these three drugs together showed comparable cell death, ~95%, *p* < 0.001, (Figure 5A) but such combination treatment did not show any advantage over GB1107 alone when treating PC-3 cells with doses of GB1107 up to 10 μM. However, increasing GB1107 to 100 μM produced ~94% cell death of PC-3 cells (Figure 5B)**.** Moreover, combination treatment in PWR-1E normal prostate epithelia cells did not show additional toxic effects other than the toxicity observed with ENZ (Figure 5C).

Further, we explored whether the combination of standard of care drug docetaxel (Doc) with GB1107 would be more effective in inducing cell death in the most aggressive CaP cell line PC-3. Our results showed that Doc alone caused greater cell death in LNCaP and LNCaP-KD than PC-3 (Figure 6A), with ~25-fold difference in their respective IC_50_ values (0.74, 0.93, and 18.93 nM respectively). However, a combination of 20 nM Doc (IC_50_ level) + 10µM GB1107 produced nearly complete cell death of PC-3 (Figure 6B). Moreover, combination 20 nM Doc (IC_50_ level) + 10 µM GB1107 showed increased toxicity in PWR-1E cells (Figure 5C). Thus, in conclusion combination of ENZ, 2-PMPA, and GB1107 is highly cytotoxic for LNCaP and LNCaP-KD cells, while Doc + GB1107 show greater efficacy in castration-resistant PC-3 cells. Overall, 2-PMPA and GB1107 demonstrate synergistic effects with ENZ and Doc for treating various CaP cell lines, without showing such effects in normal PWR-1E cells. Future studies include the evaluation of Gal-3 and PSMA inhibition in vivo and assess the utility of other inhibitors of Gal-3.

## 4. Discussion

In CaP biomarker studies, a plethora of markers such as tumor grade, volume, and stage have been suggested as useful indicators for predicting disease aggressiveness and risk stratification [26,27,28,29]. Additionally, several molecular markers such as glutathione *S*-transferase π, matrix metalloproteases, telomerase, p21, p27, cyclin D1, p53, bcl-2, E-cadherin, HER-2/neu, have also been put forward and their utility in clinical decision-making has been validated [30,31]. Despite availability of such markers, as of date there is no clear consensus on using any of them as prognostic indicators of disease progression, aggressiveness, and risk stratification. The reasons for this ranges anywhere from specificity and sensitivity of the detection methods to limited tissue availability and the concern that the inherent heterogeneity of tumor could cause false negative results [27]. With evolving research, it is now recognized that CaP is more a disease of complex interactions between CaP cells, endothelial cells, stromal cells in the tumor microenvironment, rather than as a disease of abnormally proliferating epithelial cells. The complex interactions involve numerous signaling pathways such as AR, tyrosine kinase receptor, angiogenesis, and tumor-immune escape [32]. Therefore, identifying additional markers of tumor aggressiveness and their usefulness in deciding risk stratification for treatment strategies is an urgent need. In this context “Glycomics”—a study of complete glycan structures in an organism has tremendous potential for developing novel methods of diagnosis and treatment of CaP. Similar to other cancers, CaP presents unique alterations of glycans and glycoproteins [33,34]. Glycan signatures of any cancer are among the most crucial predictors of cancer cell biological function and disease progression. Mutations in cancer cells affect both the expression of cell adhesion proteins (e.g., siglecs, galectins, and selectins) and their ligands that contain glycan chains. For example, E-selectin ligand 1 (ESL-1) controls circulating CaP cell rolling/adhesion and prostate cancer metastasis [35]. In many diseases, including aspects of the cancer progression, the patterns of cell surface glycan are altered [36]. For these reasons, determination of cell surface carbohydrate alterations in diseases has been an active subject of investigation. In search for novel biomarkers and therapeutic targets, we studied the partial glycomic signature of CaP cells by focusing on glycoproteins such as PSMA and galectins. We evaluated the expression of these glycoproteins in both CaP cell lines with distinct invasive and androgen-dependent properties and in tumor biopsy samples from treatment-free patients at different stages of the disease. We have identified a “PSMA-galectin-specific pattern” associated with CaP aggressiveness and our results highlight the significance of a combination of a Gal-3 and PSMA activity inhibition as an attractive therapeutic target in advanced stages of CaP.

In this study, LNCaP, DU-145, and PC-3 cells were independently evaluated to compare the expression of PSMA, and the different galectins belonging to galectin family. Our results in Figure 1A) show an increased expression of both PSMA and Gal-8 in LNCaP cells, while DU-145 and PC-3 showed increased expression of Gal-1 and Gal-3 and minimum expression of Gal-8 and undetectable expression of PSMA. The results on PSMA expression in different cell lines confirm and corroborate the expression reported previously [37]. However, the results on expression of different galectins in CaP cells are in agreement with certain published studies [38,39] but not with others [40]. One previous study on the expression profile of different galectins in a panel of human tumor cell lines found that all three CaP cell lines were negative for Gal-2, -4, -7, and -9 but expressed significant quantities of Gal-8 [41]. Furthermore, in the same study, Gal-1 and -3 expression was detected in DU-145 and PC-3 but not in the LNCaP cell line [41]. In contrast, another study, Laderach et al. reported significant expression of Gal-1 in LNCaP cells both at the mRNA and protein levels, albeit 20-fold lower levels than the androgen unresponsive 22Rv1 and PC-3 tumor cells [42]. Our results are more in agreement with the results of Laderach et al. The increase in fold expression of Gal-1 in the aggressive cell line DU-145 and PC-3, points to a major role for both galectins in the CaP microenvironment to promote cancer aggressiveness. Given the pleiotropic functions of Gal-1 in the tumor microenvironment, including its role in angiogenesis [43,44], cell adhesion, invasiveness [45] and immunosuppression [46,47], upregulation of Gal-1 in the aggressive cell lines will dramatically contribute to aggressive behavior of these cells. In this regard it is pertinent to point out that, Gal-1 is expressed in endothelial cells [48,49] and is upregulated in various cancer types [50]. Our findings are also consistent with the observation of increased expression of Gal-1 in aggressive and metastatic oligodendroglioma [51], aggressive B16 melanoma [44], and Kaposi’s sarcoma [43]. Additionally, we also observed increased expression of Gal-3, albeit slightly lesser than Gal-1 in our aggressive cells DU-145 and PC-3. The structure of Gal-3 allows it to oligomerize, which confers distinct Gal-3 functions in situations of both homeostasis and pathological processes, such as cancer. Gal-3 expression typically increases during cancer progression, and this expression results in both enhanced suppression of the immune response and other damaging outcomes including increased tumor progression, invasiveness, and metastatic potential. Thus, the observation of increase in fold expression of both Gal-1 and Gal-3 in the moderate and highly aggressive CaP cells contributes to the invasive/aggressive property to these two cells by providing sites for oligomerization and acting as agents for docking when the cancer cells migrate to a distant site. 

Our results on the expression of Gal-8 (Figure 1A), showing remarkable increase (~128-fold) in LNCaP cells compared to DU-145 and PC-3 cells is a novel observation. Although, confirmed expression of Gal-8 in different CaP cell line agrees with the previously reported results [52], the fold increase in the expression of Gal-8 in LNCaP cells is vastly higher than that reported in other studies [39,52]. This observation underscores the importance of Gal-8 in CaP biology and dovetails with our earlier observation that LNCaP cells are the least aggressive/metastatic variant [21]. Gal-8 originally described as PCTA-1, imparts anti-adhesive, anti-proliferative [53,54,55], and cell-cycle arrest functions [53,55,56]. In the backdrop of this knowledge, the ~128-fold increased expression of Gal-8 in LNCaP cells compared to DU-145 and PC-3 cells provide a biological function that will preclude the LNCaP cells from attaching and proliferating at distant sites. Therefore, the ~ 128-fold increase in Gal-8 expression in LNCaP cells strongly support a permissive role played by this galectin in the lack of aggressive/metastatic property demonstrated by these cells. 

The differential expression of glycoproteins, galectins, and PSMA in CaP cell lines prompted us to investigate the galectin and PSMA expression in biopsies obtained from patients with newly diagnosed untreated disease. Samples included a large spectrum of CaP Gleason scores (Table 1). We only focused on the highly expressed Gal-1 and Gal-3 and Gal-8 in the prostate biopsies. Our results show on average metastatic CaP samples expressed ~38–fold-increased levels of PSMA compared to negative controls (441.40-pixel units vs. 11.05-pixel units) and a ~ 16-fold increase in Gal-3 in metastatic CaP cells compared to negative controls (58.89-pixel units vs. 3.60). Among the CaP samples, PSMA and Gal-3 expression independently correlated weakly with Gleason score (*p* < 0.001), and the combined sum of PSMA and Gal-3 also showed a further significant (*p* < 0.001) correlation with Gleason score, whereas Gal-1 did not show any correlation with both the parameters analyzed. These data delineate a “galectin-specific signature” characterized by selective upregulation of PSMA, Gal-1 and -3 in CaP highlighting a potential role for Gal-1 and Gal-3 along with PSMA as a sensitive biomarker in advanced stages of the disease. Glycoproteins such as PSMA and galectins are involved in complex interactions in the tumor microenvironment partaking in the interactions between stromal, endothelial, and immune cell compartments. Though the role of PSMA and galectins in CaP progression and modulation has been initially ignored, current research, however, reasons that each member of the galectin family plays distinct roles in tumor cell invasiveness, inflammation, and angiogenesis [57,58]. Specifically, it has been reported that Gal-1 prevents T-cell transmigration and affects the development and survival of CaP cells [14]. Other galectins such as Gal-3 regulate CaP cell survival and homotypic and heterotypic aggregation and has been suggested as a marker of shift toward castration resistant cancerous disease [59]. Yet another galectin, Gal-8, also known PCTA1, has been shown to impact integrin-mediated cell–extracellular matrix interactions [52]. The expression of PSMA has been consistently demonstrated by IFC and other techniques in normal and hyperplastic prostate tissue, in prostatic intraepithelial neoplasia, and in invasive carcinomas [7,9]. In this study, the higher expression of PSMA measured by IFC on prostate biopsies correlated significantly with higher preoperative serum PSA, Gal-3 expression levels and high Gleason score. There are previous studies that have linked PSMA levels measured on primary prostatectomy specimens with CaP outcome, and increased PSMA expression has been associated with higher tumor grade [9], and metastatic disease [60,61]. The result of this study corroborates the results of a previously published study on PSMA levels, disease recurrence and aggressiveness in CaP. Similar to previous studies our study also used the same antibody, clone 3E6 antibody, against PSMA that recognizes the extracellular portion of the molecule in IHC with great success [62,63]. Despite the current interest in PSMA and galectins in CaP the clinical use of combined PSMA and galectin expression in primary metastatic CaP has not been evaluated previously as a biomarker for aggressive CaP diagnosis. In our study apart from observing a correlation with PSMA and Gleason Score we also observed an increased correlation with the expression levels of PSMA and Gal-3. Further when the combined index (PSMA + Gal-3) is taken, we observed better stronger correlation with Gleason score. Such an observation is the first report on evaluation of the combination of PSMA and galectin expression status as a predictor of CaP disease stage and aggressiveness. In the process of identifying this we uncovered a “PSMA-galectin mark” linked to prostate cancer aggressiveness while looking for new biomarkers and therapeutic options. From a translational viewpoint, the combined index of PSMA + Gal-3 can be used as a tissue and serum prognostic biomarker for discriminating clinically insignificant CaP from more aggressive and higher stage tumors. Other potential applications include using serum levels of this index to monitor the response to treatment and potentially as a therapeutic target for CaP. The combined sum index can thus be used as a novel marker to complement already existing diagnostic markers for CaP. 

Castration-resistant prostate cancer treatment is challenging and incurable. Therapeutic options for managing metastatic castration-resistant prostate cancer are limited. Our study establishes a correlation between functionally important cell surface glycoproteins PSMA and Gal-3 and metastatic prostate cancer, as well as their expression in CaP cell lines of varying metastatic potential. Our objective apart from future exploitation of these tumor markers in disease diagnosis was also to see if these two functionally important proteins can be targeted for therapy. Novel therapeutic approached to combat castration resistant prostate cancer has relied on combination of inhibitors along with AR inhibition as an essential step to elicit therapeutic efficacy in [64]. Such studies have also shown that inhibitors of PI3K, Akt, or mTOR used along with AR inhibition have limitations, including off-target effects caused by the constitutive expression of these proteins in other organs. Under such a scenario we reasoned that inhibition of PSMA, and Gal-3, biologically active glycoproteins expressed only in metastatic castration variant can be an alternative therapeutic approach. Toward this objective we designed experiments to assess the effectiveness of PSMA inhibitor (2-PMPA) and Gal-3 inhibitor (GB1107) individually and along with enzalutamide (AR-Inhibitor) or Docetaxel (microtubule stabilizer) for further studies. 2-PMPA and GB1107 have shown effectiveness in inhibiting their respected targets, however the usefulness of their application conjointly in the setting of enzalutamide remains to be seen. Our results show enzalutamide inducing dose-dependent cytotoxicity in AR^+^ and PSMA^+^ LNCaP cells, AR^+^ and PSMA^−^ LNCaP KD cells and AR^+^ and PSMA^−^ DU-145 cells but not in AR^−^ and PSMA^−^ PC-3 cells Figure 4A). This response is largely tracked to the known mechanism of action of enzalutamide and the AR expression status. The responsive cells LNCaP and LNCAP-KD express high levels of AR protein expression and non-responsive PC-3 cells have no detectable levels of AR expression [65,66]. Enzalutamide is a second generation, non-steroidal anti-androgen drug that acts by binding to AR and preventing its translocation to the nucleus for growth-related signal transduction [67]. Trials have demonstrated strong efficacy of this drug in improving time to progression and extending overall survival in patients whose tumor is AR responsive [68]. Moreover, in responsive cells the doses required to inhibit the growth were in the µM range, which also conforms to the previously reported observation [69]. Importantly, the observation that PC-3 cells did not show any response to enzalutamide suggests that apart from being AR null, these cells have alternative survival and signal transduction mechanism(s) present to mediate AR effects [70]. Together, these data are consistent with the possibility that AR activity may be impacted in these cell lines when treated with high doses of enzalutamide. Our results also showed higher expression of functionally active glycoproteins PSMA and Gal-3 in prostate cancer patients so, we next evaluated if targeting of PSMA and Gal-3 independently could serve as potential pharmacological treatment avenue. Our results show that treatment of CaP cells with 2-PMPA alone shows a dose-dependent effect only in PSMA-positive cells LNCaP and not in PSMA-negative LNCAP-KD, DU-145, and PC-3 cells (Figure 4B). These results are consistent with the known role of 2-PMPA in inhibiting the enzymatic activity of PSMA [71]. The lack of response in the other three cells likely reflects the lack of PSMA activity in these three cells [37]. These results are also on the lines of observation by Kaittanis et al. [72]. With the known role of PSMA in activating the PI3K pathways and negatively regulating AR pathway, our results have significant clinical implications, because PSMA can be used in tandem with antiandrogens, such as enzalutamide to block both the PSMA activity and AR pathway simultaneously.

In our studies with Gal-3 inhibitor (GB-1107) when used independently, we observed dose-dependent cytotoxicity in all three cell lines with LNCaP and DU-145 more susceptible than the PC-3 cells (Figure 4C). This result strongly suggests that the level of cytotoxicity we observed roughly parallels the Gal-3 expression level in these cells and to obtain the maximum response in the high Gal-3 expressing cells we need to titrate the doses further to obtain complete growth reduction. However, the combination treatment studies with enzalutamide, 2-PMPA, and Gal-3 inhibitor (Figure 5A,B) is novel, and the observation suggests that combination treatment has a great potential for treating PSMA and Gal-3 expressing prostate cells compared to using each individually. Nevertheless, in the PSMA and AR-negative prostate cells PC-3, the combination treatment with Doc and Gal-3 inhibitor seems more effective (Figure 6A,B). Although Doc is the drug of choice for treatment of castration-resistant prostate cancer, despite early success, resistance to Doc sets in and patients are insensitive to treatment with Doc at this stage. Subsequent to this, considerable effort is being directed at applying potential benefits of Doc-based combination therapy to the clinic, aiming to increase the number and/or duration of responses. Preclinically, multiple classes of agents with other mechanisms of action have shown an additive or synergistic activity however, so far, no such therapeutic combination has shown a survival improvement in vivo when added to Doc. The results of this study showing a strong correlation between Gal-3 expression and tumor aggressiveness prompted us to study the combination of Gal-3 inhibition with Doc as a remedy. The results of our study on the combination of Gal-3 inhibition and Doc showing 94% cytotoxicity provide a glimmer of hope to the Doc-resistant patients, and present a new combination to overcome Doc resistance.

## 5. Conclusions

In conclusion, our data show that overexpression of PSMA and Gal-3 in primary prostate cancer correlates with other traditional adverse prognostic factors and independently predicts high-grade tumors. Additionally, the combined index of PSMA and Gal-3 is a predictive biomarker of CaP aggressiveness and prognostic indicator of therapy failure with standard treatment. Combination of ENZ, 2-PMPA, and GB1107 is highly cytotoxic for LNCaP and LNCaP-KD cells, while DOC + GB1107 show greater efficacy in castration-resistant PC-3 cells. Overall, 2-PMPA and GB1107 demonstrate synergistic effects with ENZ and DOC for treating various CaP cell lines. Future studies include the evaluation of Gal-3 and PSMA inhibition in vivo and assess the utility of other inhibitors of Gal-3.

## Figures and Tables

**Figure 1 cancers-14-02704-f001:**
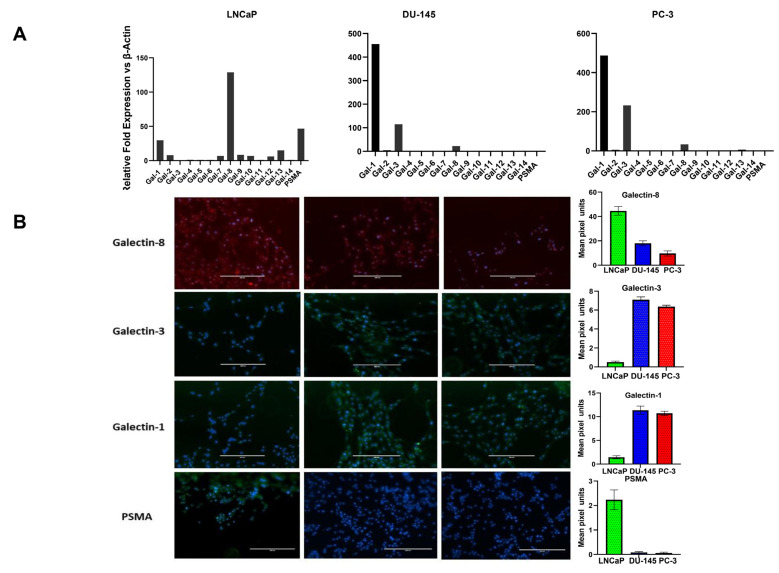
Expression profile of different galectins and PSMA in prostate cancer cell lines varying in aggressive potential and AR expression status. (**A**) Transcriptional profile of galectins and PSMA by Q-PCR. Results are expressed as galectin or PSMA mRNA relative to β-Actin. Cell line data are shown according to their stated aggressive potential and AR expression status. Left, minimally aggressive, Androgen receptor^+^, LNCaP cells; middle, moderately aggressive, AR^+^ DU-145 cells; right, the most aggressive AR^−^, PC-3 cells. Data are expressed as mean ± SD of 4 independent experiments. (**B**) immunocytochemical analysis of galectins and PSMA in prostate cancer cells of varying aggressive potential adhered onto poly-l-lysine–coated glasses (magnification, ×40).

**Figure 2 cancers-14-02704-f002:**
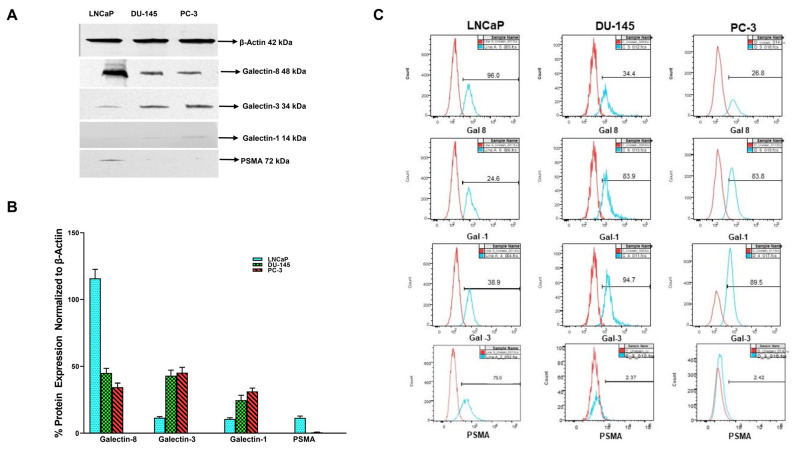
Basal expression of different galectins and PSMA in prostate cancer cell lines of varying aggressive potential by Western blot and flow cytometry analysis. (**A**) Representative Western blot images (Appendix A) showing differential expression of galectins and PSMA in prostate cancer cells probed by specific antibodies for galectins and PSMA. Cells were lysed and equal amount of lysates (10 µg) were loaded/lane, run on 10% SDS reducing gel electrophoresis, transferred to PVDF membranes for Western blotting for β-actin, galectins, and PSMA as described in the Section 2. Representative Western blots from three separate experiments yielded similar results. (**B**) Quantitative values of densitometric analysis of scans of Western blots normalized to β-actin and expressed as a percentage of β-Actin expression in each cell line. (**C**) Representative histograms of our flow cytometry data showing expression of galectin-8 (top panel), galectin-1 (second panel), galectin-3 (third panel), and PSMA (bottom panel) in LNCaP, DU-145, and PC-3 cells respectively. Unstained cells of each cell line serve as a control when overlayed onto the stained population and allows identification of the galectin-8, galectin-1, galectin-3, and PSMA expressing cells. *Y*-axis shows cell counts and *X*-axis shows FITC positive cells.

**Figure 3 cancers-14-02704-f003:**
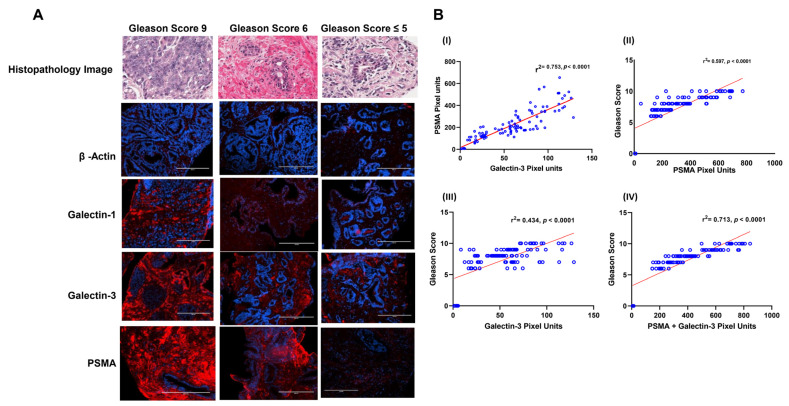
Immunofluorescence image intensity and correlational analysis of β-actin, galectin-1, galectin-3 and PSMA in prostate cancer biopsies of varying Gleason score as quantitated by image J Analysis and numerical data on fluorescence intensity analyzed using GraphPad Prism. (**A**) Representative immunofluorescence images of different Gleason Score biopsied samples. Immunofluorescence staining was done using specific primary antibody as described under the Section 2. The secondary antibody used had Alexa Fluor 594 (Red) and the nucleus was stained with DAPI (blue) in all cases. The expression levels of each were quantitated based on the intensity of the fluorescent signal analyzed using the computer image analysis image J software (National Institutes of Health, Bethesda, MA, USA). Standard immunofluorescence staining procedures as described under the Section 2 were followed. Imaging was performed with the EVOS^®^ FL Cell Imaging System and images shown are representative images for proteins probed with Gleason Score 9 (left panel), Gleason score 6 (middle panel) and no Gleason Score (right Panel 3). (**B**) Correlational analysis of numerical data of staining intensity of (I) PSMA pixel units vs. Gal-3-pixel units, (II) PSMA pixel units vs. Gleason Score, (III) galectin-3 pixel units vs. Gleason score and (IV) PSMA + galectin-3 pixel units vs. Gleason score.

**Figure 4 cancers-14-02704-f004:**
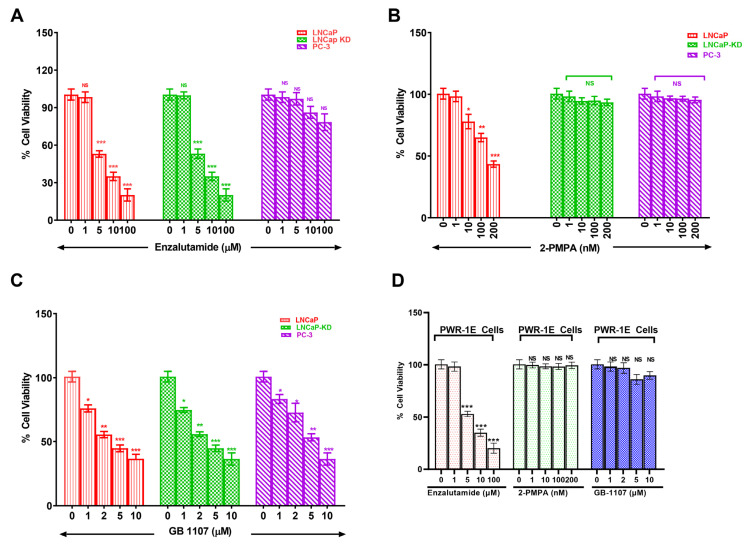
Dose dependent cytotoxicity of Enzalutamide (**A**), 2-Phosphonomethyl pentanedioic acid (2-PMPA) (**B**), GB 1107 (**C**) in CaP cells of varying aggressive potential and of all these drugs on PWR-1E cells (**D**). Prostate cancer cells of varying aggressive potential LNCaP(low), LNCaP-KD(low), PC-3(high) cells and normal prostate epithelial cells PER-1E cells were treated with indicated concentrations of either enzalutamide, 2-PMPA or GB-1107 for 72 h following which viability of surviving cells was determined by the 3 (4,5 dimethylthiazol 2 Yl) 2,5 diphenyltetrazolium bromide (MTT) assay using a microtiter plate reader (Bio Tek Instruments, Inc., Winooski, VT, USA) at 570 nm. Results are the mean ± SD of four independent experiments done in triplicate. Statistical significance: * *p* < 0.05, ** *p* < 0.01, *** *p* < 0.001, NS = not significant.

**Figure 5 cancers-14-02704-f005:**
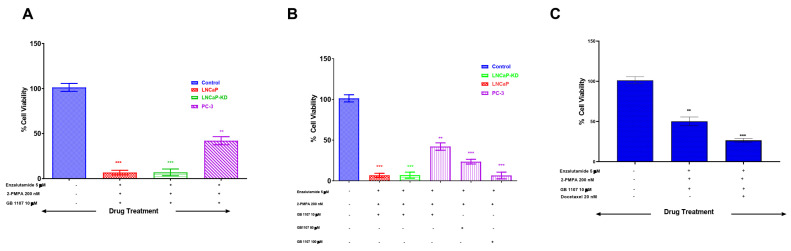
Combination treatment of prostate cancer cells and normal prostate epithelial cells with different chemotherapy drugs. LNCaP, LNCaP KD, PC-3, and PER-1E cells were treated with indicated doses of ENZ, ENZ+ 2-PMPA, or ENZ+ 2-PMA+GB-1107 for 72 h and were analyzed for viability of surviving cells as determined by the 3 (4,5 dimethylthiazol 2 Yl) 2,5 diphenyltetrazolium bromide (MTT) assay. (**A**) Combination treatment of all three drugs at reported optimal doses in different prostate cancer cells. (**B**) Combination treatment of ENZ and 2-PMPA at reported optimal doses and increasing concentration of GB-1107 in different prostate cancer cells, and (**C**) combination treatment of ENZ and 2-PMPA and GB-1107 at reported optimal doses with or without Docetaxel in normal prostate epithelial cells, PWR-1E. Results are the mean ± SD of four independent experiments done in triplicate. Statistical significance: ** *p* < 0.01, *** *p* < 0.001.

**Figure 6 cancers-14-02704-f006:**
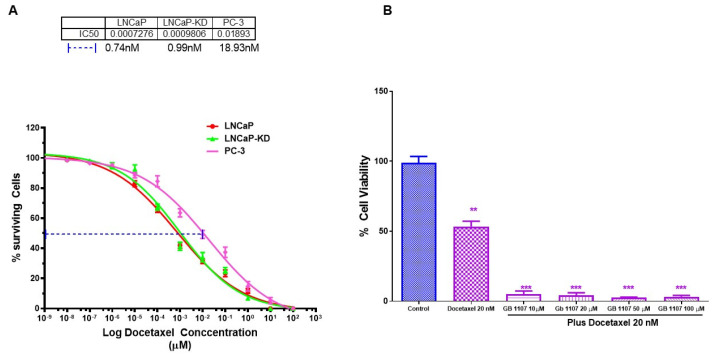
Combination treatment of prostate cancer cells with varying aggressive potential with Docetaxel and GB 1107. LNCaP, LNCaP KD, and PC-3 cell with varying aggressive potential were treated with indicated doses of Docetaxel for 72 h, and analyzed for viability of surviving cells as determined by the 3 (4,5 dimethylthiazol 2 Yl) 2,5 diphenyltetrazolium bromide (MTT) assay. (**A**) Dose effect and IC _50_ for Docetaxel, (**B**) combination of IC_50_ level of Docetaxel with varying concentration of GB 1107 on viability of PC-3 cells as determination by 3 (4,5 dimethylthiazol 2 Yl) 2,5 diphenyltetrazolium bromide (MTT) assay. Results are the mean ± SD of four individual experiments done in triplicate. Statistical significance: ** *p* < 0.01, *** *p* < 0.001.

**Table 1 cancers-14-02704-t001:** Clinical parameters, galectin-1, galctin-3, PSMA expression status in 115 cases of untreated cases of prostate cancer.

Gleason Score	Age in (Years)	Number of Patients	Serum PSA at Diagnosis (ng/mL)	PSMA Intensity (Pixel Units)	Galectin-1 Intensity (Pixel Units)	Galectin-3 Intensity (Pixel Units)	PSMA + Galectin-3 Intensity (Pixel Units)
≤5	71	10	3.25 ± 0.56	7.45 ± 1.53	6.75 ± 5.07	3.60 ± 1.23	11.05 ± 2.45
6	73	9	5.76 ± 2.91	153.40 ± 26.71	16.79 ± 4.55	33.40 ± 21.52	194.52 ± 37.47
7	77	18	47.49 ± 16.70	196.91 ± 40.14	13.12 ± 5.36	59.34 ± 32.46	270.54 ± 61.33
8	72	41	110.32 ± 23.39	291.17 ± 109.17	12.85 ± 7.84	48.18 ± 13.42	390.67 ± 70.90
9	72	30	324.46 ± 92.92	481.26 ± 118.61	17.12 ±11.68	64.03 ± 25.97	586.91 ± 88.93
10	74	7	431.24 ± 81.51	591.49 ± 132.75	14.89 ± 14.93	92.98 ± 16.85	709.47 ± 86.38

## Data Availability

The data presented in this study are available on request from the corresponding author.

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
