# Peer review of "Use of Glycoproteins—Prostate-Specific Membrane Antigen and Galectin-3 as Primary Tumor Markers and Therapeutic Targets in the Management of Metastatic Prostate Cancer"

_cancers, 2022, doi:10.3390/cancers14112704_

Round 1
Reviewer 1 Report
Prof. Aalinkeel has designed a study about specific markers in prostate cancer, which can be useful in further cancer diagnosis and finding therapeutic agents for treatment. Prostate cancer samples were separated into various grades; low, moderate and aggressive in cell line, and using Gleason Score in the separation of biopsy samples. Various markers were compared in samples of different grades. Various anti-cancer agents were also used individually or in combination to evaluate their effectiveness in different cancer cells. The article is very interesting with good novelty. With proper adjustment by adding some more data, I believe the article will have a good chance to be accepted.
My comments are listed below:
- First, I am not sure is it an issue from the submitting system or an issue from the authors, the image quality of most figures in the article needs major improvement. The front size inside the figures and the resolution of the figures are way too small and low. Figure 1B, 2, 3, 4B, 5B are barely readable even on large screen. Please talk with the editorial board if the issue is from the submitting system, or please improve the quality and resolution of the figures.
- I highly suggest that western blot results in cell line should be added into figure 1. Especially when the result of PSMA do not correlate with biopsy sample in figure 2. In figure 1, PSMA is expressed in low aggressive LNCaP cell lines but not expressed in high aggressive DU-145 and PC-3 cell line. However, in figure 2, the higher the Gleason Score the higher PSMA protein was found. Also in Figure 3B, 2-PMPA shown good toxicity in LNCaP cell. I doubt that the PCR results in Figure 1 may not be totally correct and should re-confirmed with western blot demonstrating that PSMA gene did indeed transferred to PSMA protein. Figure 1B is unreadable on my side so I cannot judge does the protein staining results match with PCR in Figure 1A. Still, I feel that using western blot for protein amount analysis is needed.
- I do not feel that using immunofluorescence image analysis by image J is a good method to quantify protein amount in biopsy samples. I also feel that the correlation between PSMA+ Galectin-3 and Gleason score is not that significant. I doubt that there will be significant correlation after statistical analysis. So I suggest that the author should use more appropriate instrument, such as flow cytometry, in quantify protein concentration.
- The combination treatment results of prostate cancer cells is very promising and significant. However, I still suggest it will be better if the author can also evaluate the toxicity of combination treatment in normal cells. Combining different anti-cancer agents will have better toxicity in cancer cells, this is no surprise. I believe the readers will be more interested to know if they will also deal as much toxicity as in normal cells.
Author Response
Reviewer 1 Comment 1: "First, I am not sure is it an issue from the submitting system or an issue from the authors, ………..., or please improve the quality and resolution of the figures".
Response: We thank the reviewer for pointing this. This problem with the resolution was the smaller size images loaded by the authors. As the web-based system of submission was a lot perplexing, we are talking with the editorial board to submit the figures on the web. The revised figures attached here with are all being submitted as individual high-resolution images and we are talking to the editorial board to resolve the issue.
Reviewer 1 Comment 2: "I highly suggest that western blot results in cell line …………..for protein amount analysis is needed"
Response: We appreciate the reviewer comment and recognize the importance the western blot. For cell culture experiments, we have now added western blot experiments and flow cytometry data. Figure 2A,-C are added to the document. Figure 1B, which is unreadable, is now given as a larger sized figure. The data from these two additional experiments confirm our observations from the Gene expression study. We appreciate the Reviewers' comment on LNCaP cells vs biopsy samples. However, the difference in the expression of PSAM vs Glaectin-3 in cells in the biopsy samples may be due to the inherent difference in difference in vitro vs in vivo condition.
Reviewer 1 Comment 3: I do not feel that using immunofluorescence image analysis…..the author should use a more appropriate instrument, such as flow cytometry, to quantify protein concentration
Response: The correlation between PSMA and Galectin was analyzed statistically and there was a high level of statistical significance and correlation between samples analyzed. These data are now included in the figure as figure 3B(I-IV). Also, since the samples available with us are precious histology blocks, performing flow cytometry on such samples is not technically possible and we hope the reviewer understands our limitations with regard to flow cytometry on biopsy samples.
Reviewer 1 Comment 4: The combination treatment results of prostate cancer cells are very promising and significant…. I believe the readers will be more interested to know if they will also deal as much toxicity as in normal cells.
Response: We appreciate the reviewer for acknowledging the significance of the combination treatment study. We agree with the reviewer and performed experiments of the combination treatment on PWR-1E normal prostate epithelial cells and give our results in figure 4D and 5C. As mentioned by the reviewer, there was a dose-dependent toxicity in the normal prostate epithelial cells when treated with Enzalutamide and Docetaxel however such toxicity was noted when the cells were treated with GB-1107 or 2-PMPA, this result is now added in the results section on page 10 line 16-20.
Thank you for your consideration of this manuscript and hopefully the concerns are addressed in our revised version.
Reviewer 2 Report
This well-written paper by Satish Sharma, et al., entitled “Use of glycoproteins- prostate specific membrane antigen and galectin-3 as primary tumor markers and therapeutic targets in the management of metastatic prostate cancer”. In the initial steps of reviewing this manuscript, an extensive literature search was undertaken and concluded that the authors introductory statement was correct in that there was no published data on the role played by the co-existence of altered prostate specific membrane antigen and/or galectin expression in different stages of prostate cancer with regards to the cancers progression or metastatic character. Addressing this, and employing the significantly variable aggressive phenotypic prostate cell lines of PC‐3, DU-145, and LNCaP, this study clearly demonstrates that prostate cancer prostate specific membrane antigen and galectin-3 overexpression correlated well with traditional prognostic factors that use current well-known prostate cancer biomarkers. The evaluation of these two cell surface tumor markers, taken together, as studied with these cell lines, provide significantly to a knowledge-base upon which a therapeutic approach can be better developed.
Reviewer 3 Report
Authors should be congratulated for their work. The topic is really challenging and it will be interesting to understand how this markers could be employed to better improve survival outcome in patients with lymphovascular invasion as well as mPCA patients. The manuscript is easily readable. The figures are clear and tables are well organized. The article is suitable for a publication after major revision. Indeed, Authors should clarify several points:
- Are any data available on the definitive histology? Could the Galectins and PSMA improve risk stratification of PCA, specifically for the GS 7? Are the Galectins and PSMA correlate with upgrading or upstaging of PCA?
- Are any data available of the role of Galectin to recognize rare tumor patterns such as glomeruloid, mucinous or cribriform variant?
Author Response
Reviewer 3 commnet1: Are any data available on the definitive histology? Could the Galectins and PSMA improve risk stratification of PCA, specifically for the GS 7? Are the Galectins and PSMA correlate with upgrading or upstaging of PCA?
Response: R: The authors believe that the expression of Galectins and PSMA will improve risk stratification of PCA, specifically for the GS 7 and above. However, at this time we are unable to predict if it can be used for upgrading or upstaging of PCA .
Reviewer 3 comment 2: Are any data available of the role of Galectin in recognizing rare tumor patterns such as glomeruloid, mucinous, or cribriform variant?
Response: After an extensive review of the literature, the authors did not observe any reports of the role of galectins in recognizing rare tumor patterns such as glomeruloid, mucinous and cribriform variants.
Round 2
Reviewer 1 Report
Prof. Aalinkeel had made great improvement in the manuscript. Previously I requested more results on western blot, flow cytometry and etc. Prof. Aalinkeel had provide good results on every one of them. Although I still have minor issue on the quality of the figures as the author claim that high-resolution images were submitted individually; I cannot find where to download them. Still I believe this should be an issue of the submission system and should not affect the overall quality of the article. So I believe the article has been largely improved after revise; after minor adjustment on the figures quality with the editorial board, the article is ready for publish.
Reviewer 3 Report
Authors should be congratulated for their work. The topic is really challenging and it will be interesting to understand how this markers could be employed to better improve survival outcome in patients with lymphovascular invasion as well as mPCA patients. The manuscript is easily readable. The figures are clear and tables are well organized. Authors improved the quality of the manuscript.The article is suitable for publication.